# PreCo: Enhancing Generalization in Co-Design of Modular Soft Robots via Brain-Body Pre-Training

**Yuxing Wang[1], Shuang Wu[2], Tiantian Zhang[1], Yongzhe Chang[1*], Haobo Fu[2], Qiang Fu[2], Xueqian Wang[1*]**

[1]Tsinghua University [2]Tencent AI Lab

**Abstract:** Brain-body co-design, which involves the collaborative design of control strategies and morphologies, has emerged as a promising approach to enhance a robot's adaptability to its environment. However, the conventional co-design process often starts from scratch, lacking the utilization of prior knowledge. This can result in time-consuming and costly endeavors. In this paper, we present PreCo, a novel methodology that efficiently integrates brain-body pre-training into the co-design process of modular soft robots. PreCo is based on the insight of embedding co-design principles into models, achieved by pre-training a universal co-design policy on a diverse set of tasks. This pre-trained co-designer is utilized to generate initial designs and control policies, which are then fine-tuned for specific co-design tasks. Through experiments on a modular soft robot system, our method demonstrates zero-shot generalization to unseen co-design tasks, facilitating few-shot adaptation while significantly reducing the number of policy iterations required. Our video is available here.

**Keywords:** Co-design, Pre-training, Modular Soft Robots

## 1 Introduction

Nature does not treat the development of the brain and body as separate processes, indicating that cognitive processes are intricately connected to the body and the external environment in which organisms operate [1, 2]. This theory holds significant implications for the robotics community. To enable effective interaction with the environment, it is essential to prioritize the co-design of both physical bodies and control systems of robots. In this work, we consider the co-design of Modular Soft Robots (MSRs), which are a promising category of flexible robotic systems that offer designers the ability to construct robot bodies by combining various types of deformable cubes, and the control signals can be generated by adjusting cubes' volume (Figure 1). Currently, the majority of related co-design studies [3, 4, 5] for MSRs mainly focus on "one robot one task", where the primary objective is to discover the optimal robot morphology and controller for a specific task. However, this approach seems to diverge from biological morphologies, such as the human body, which inherently possesses the ability to perform multiple tasks.

As a matter of fact, even improving a robot morphology for a single task can be highly challenging due to: (1) the presence of a severe combinatorial explosion within the robot design space and (2) the existence of incompatible state-action spaces that necessitate training a separate control policy for each morphology. Consequently, past studies [6, 7, 8, 9, 10] often address these challenges by considering the evolution of body and control as separate processes, directly conducted within the large, high-dimensional design space. In other words, these methods typically learn from scratch, neglecting prior co-design knowledge, resulting in costly and inefficient endeavors. But how can we leverage this knowledge to enhance the co-design process for new applications?

---

*Correspondence to: Yongzhe Chang and Xueqian Wang {changyongzhe, wang.xq}@sz.tsinghua.edu.cn

7th Conference on Robot Learning (CoRL 2023), Atlanta, USA.

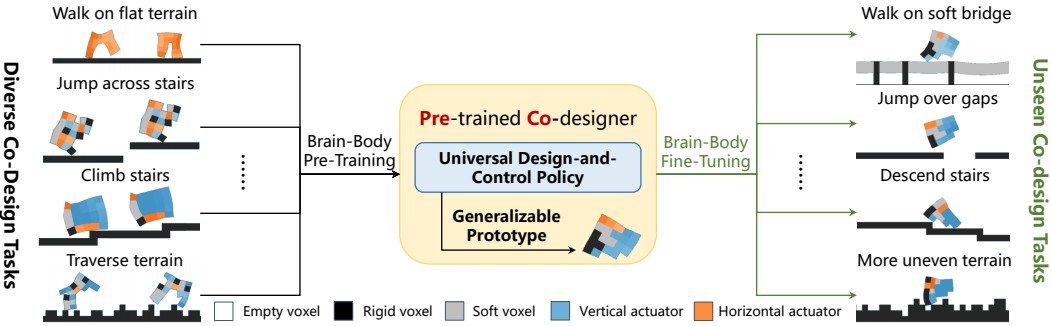

Figure 1: Workflow of PreCo. At the heart of PreCo lies a universal co-design policy, which undergoes pre-training using end-to-end deep reinforcement learning on a diverse set of co-design tasks. The resulting pre-trained co-designer is utilized to generate initial designs and control policies, which are further fine-tuned for unseen tasks.

In this paper, we propose PreCo, a methodology that entails pre-training a universal co-design policy to grasp the interdependencies between the robot morphology, control and tasks. Following this foundational step, the policy is utilized to generate initial designs and control strategies, which are then fine-tuned for unseen co-design tasks, thereby reducing the learning burden. Precisely, our approach implies that both the morphologies and control strategies of a robot stem from the same set of parameters. Any mutation to the parameter simultaneously affects the two, and they are trained using deep reinforcement learning. In contrast to conventional co-design methods that segregate not only the optimization processes of the brain and body but also the parameters that generate them, we draw inspiration from pleiotropy in biology, which refers to a single gene expressing multiple phenotypic traits [11, 12, 13]. Thus, our method eliminates the need for a robot population and provides empirical evidence of its capacity to enhance the efficiency of the learning process.

Our study offers the following key contributions: Firstly, we introduce brain-body pre-training. Subsequently, we present PreCo, a novel approach that learns a universal design-and-control policy capable of handling multiple challenging co-design tasks for modular soft robots. Secondly, using the pre-trained co-design policy, we showcase that properly integrating prior knowledge makes co-design on new tasks easier in several ways: enabling improved sample efficiency, zero-shot generalization and effective few-shot adaptation, providing the benefit over training from scratch. Thirdly, through empirical analysis, we demonstrate that the shared policy structure of PreCo exhibits greater robustness in terms of mitigating premature convergence, resulting in improved exploration and flexibility. Furthermore, our work provides the first experimental comparison among meta-learning, curriculum learning and pre-training methods in addressing co-design problems.

## 2 Related Work

**Robot Co-Design** The process of developing both the physical body and the cognitive capabilities in nature is intricately intertwined [14, 15]. To replicate this fundamental principle, robot designers are tasked with concurrently optimizing the morphology and control strategy of robots. In the field of Evolutionary Robotics (ER), researchers have extensively investigated the application of Evolutionary Algorithms (EAs) for co-designing robotic systems [6, 8, 10, 16]. A prominent focus in this field lies in the representation of robot morphologies. Various approaches, such as generative encoding schemes [17, 18], have been explored to facilitate the discovery of novel and efficient designs. Techniques like neural networks [19, 20], Neural Cellular Automata (NCA) [21, 22, 23] and Compositional Pattern-Producing Networks (CPPNs) [4, 24] have been employed to generate diverse and complex robot morphologies, enabling the exploration of a broad design space. In addition, EAs can also be integrated with Reinforcement Learning (RL), allowing robots to evolve and improve their behaviors through interactions with the environment [3, 25]. These approaches, however, adopt

separate parameters for generating the robot morphology and control, which are optimized in a bi-level fashion. Consequently, they rely on a population of design prototypes to facilitate exploration, leading to challenges in terms of sample efficiency and computational requirements. In contrast to these population-based approaches, we propose an alternative methodology that utilizes a universal policy representation, enabling the robot morphology and control strategy to be derived from the same set of parameters and jointly optimized. Through empirical experiments, we demonstrate that this shared representation facilitates the exploration of the design space, leading to enhanced sample efficiency and increased flexibility.

In addition to EA methods, when we have access to certain aspects of the system's physical dynamics, a model-based differentiable simulator can be employed to jointly optimize the design parameters and control using Back-Propagation Through Time (BPTT) [26, 27, 28, 29, 30, 31]. In contrast, our work specifically addresses the model-free setting, where system modeling is not required. To achieve this, Policy Gradient (PG) methods can be used to approximate the gradient of design parameters or evaluate the fitness of a robot morphology [5, 32, 33, 34, 35]. Building upon this technology, PreCo is introduced as a novel approach that distinguishes itself from previous works by tackling a more challenging objective of addressing multiple co-design tasks simultaneously.

Many endeavors have been conducted to bring robot co-design to real-world settings, including co-designing soft hands [36], voxel-based soft robots [37, 38], soft robotic fishes [39, 40] and soft legged robots [41]. To enable effective sim-to-real transfer, numerical mathematic techniques like Finite Element Method (FEM) [42] or Material Point Method (MPM) [43] together with a high-quality simulator are required to model and simulate soft-body physics. Besides, factors like material imperfections, air resistance, friction and many others come into play, iteratively refining the design and control algorithms based on real-world feedback is also needed.

**Multi-task Reinforcement Learning** In this study, we approach the challenge of brain-body pre-training by adopting a Multi-task Reinforcement Learning (MTRL) framework, which has garnered considerable interest in the field of embodied intelligence [44, 45, 46, 47, 48]. MTRL involves the training of an agent to perform multiple tasks concurrently, aiming to leverage shared knowledge across tasks for improved learning efficiency and generalization. However, previous research based on the transformer structure primarily focuses on training a universal controller for multiple robot bodies [9, 49, 50, 51, 52, 53]. PreCo takes a further step by exploring how the intrinsic brain-body connections can be utilized to improve efficiency and generalization when facing new applications. In essence, our method aims to avoid learning from scratch, sharing a similar spirit with curriculum learning [5, 19, 54] and meta-learning [25, 55, 56, 57] but differs in its framework.

## 3 Preliminaries

**Reinforcement Learning** In our study, we approach the problem of brain-body pre-training for a set of $K$ co-design tasks by formulating it as a MTRL problem. In the domain of RL, the problem is typically formulated as a Markov Decision Process (MDP), defined by a 5-tuple $(\mathcal{S}, \mathcal{A}, P, r, \gamma)$. Here, $\mathcal{S}$ represents the state space and $\mathcal{A}$ represents the action space. The transition function $P : \mathcal{S} \times \mathcal{S} \times \mathcal{A} \to [0, 1]$ determines the probability of transitioning from one state to another given a specific action. The reward function $r(s, a) : \mathcal{S} \times \mathcal{A} \to \mathbb{R}$ assigns a numeric value to the state-action pairs, indicating the desirability of taking a particular action in a given state, and the discount factor $\gamma \in (0, 1]$ specifies the degree to which rewards are discounted over time. Our goal is to find policy parameters $\theta$ which can maximize the average expected reward across all co-design tasks: $\frac{1}{K} \sum_{k=1}^{K} \sum_{t=0}^{\infty} \gamma^t r_t^k(s_t, a_t)$, here the policy is represented by a deep neural network parameterized as $\pi_\theta(a_t|s_t)$, which maps from states to distributions over actions.

We employ Proximal Policy Optimization (PPO) [58], a popular RL algorithm that is widely used in a variety of robot tasks. The algorithm utilizes a surrogate objective function that approximates the policy gradient, and the objective function of PPO is:

$$J(\theta) = \mathbb{E}_t \left[ \frac{\pi_\theta(a_t|s_t)}{\pi_{\theta_{old}}(a_t|s_t)} \hat{A}_t - \beta D_{KL}(\pi_{\theta_{old}}(\cdot|s_t) \| \pi_\theta(\cdot|s_t)) \right] \tag{1}$$

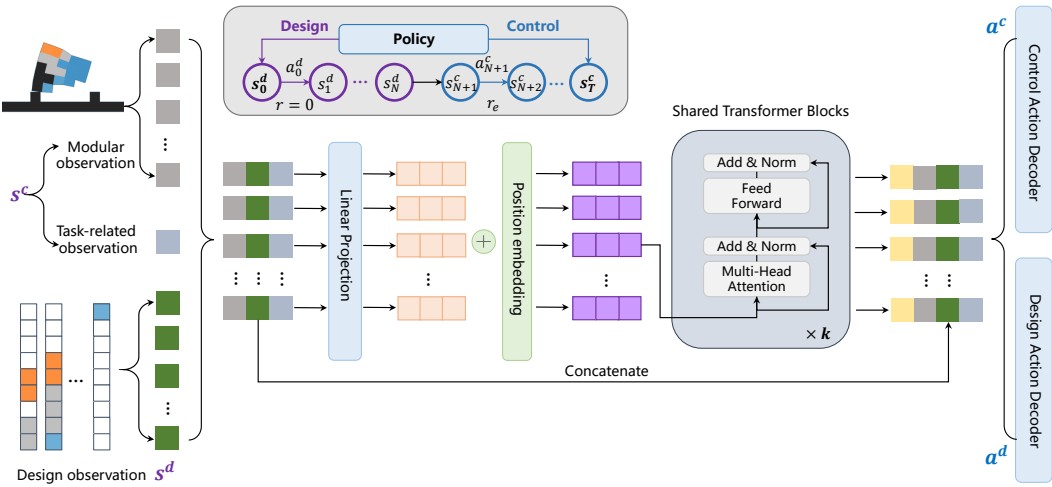

Figure 2: Architecture of the co-design policy. Our policy is designed with a shared structure that influences both morphology and control. It receives unified design-and-control observations and generates corresponding actions. This policy operates under the framework of reinforcement learning, where the design and control processes are unified as a single MDP (gray box).

where $\hat{A}_t$ is the advantage estimation, $\mathbb{E}_t$ represents the empirical average over a batch of generated samples. By iteratively collecting experiences and optimizing $J(\theta)$, the policy $\pi_\theta(a_t|s_t)$ is updated in the direction of maximizing the cumulative reward.

**Transformer** Transformer [59] is a popular neural network architecture that has revolutionized the domain of natural language processing and computer vision [60, 61, 62, 63, 64], and has become a fundamental component in many cutting-edge models. At its core, transformer employs a powerful self-attention mechanism to capture the dependencies and relationships among elements in a sequence. It allows the model to dynamically allocate attention to different parts of the input sequence based on their relevance. In each self-attention layer, attention weights are computed for each element by considering its interactions with all other elements in the sequence. These weights play a crucial role in aggregating information from the entire sequence, enabling the transformer to generate comprehensive and informative representations for each element.

The transformer architecture is well-suited for modular robot systems because it is agnostic to incompatible state-action spaces. In our work, we model the co-design of modular soft robots as a sequence-to-sequence task. Under this framework, the local observations from all voxels are organized in sequences. By leveraging the self-attention mechanism, the co-design policy can focus more on crucial parts of the state space and capture the internal dependencies between voxels, allowing the policy to dynamically adjust its focus depending on the input context, which caters to the need for dynamically accommodating changes in morphologies.

## 4 Brain-Body Pre-Training

Our motivation for employing brain-body pre-training is that given the assumption of the existence of underlying structural similarity between the pre-training tasks and the target tasks, properly integrating prior co-design knowledge into a universal co-design policy makes robot co-design easier. For instance, if a target task requires the robot to master a complex skill, such as traversing across extremely uneven terrain, this skill can be broken down into some foundational abilities like walking, ascending stairs and surmounting minor obstacles. During pre-training, the universal co-design policy aims to extract basic brain-body links from these tasks and merge them. When facing specific target tasks, it is anticipated to leverage the prior knowledge, thereby alleviating the co-design

challenge. In the remaining section, we describe details about our universal co-design policy, which is optimized end-to-end through reinforcement learning within a unified state-action space.

**Universal Co-Design Policy** We start by reviewing co-design methods [5, 35] that utilize RL to approximate the gradient of design and control parameters. At the start of each episode, a dedicated design policy takes a finite number of actions in order to develop a robot morphology, and no reward is assigned to the design policy during this period. Subsequently, the resulting robot is consumed by a control policy to collect the environmental rewards, which also provides learning signals for the design actions. Using the RL method, two policies are optimized jointly to maximize the performance for the given task.

However, a notable issue of this approach is the presence of an imbalanced sample distribution between design and control. While this imbalance might not be immediately evident during the initial stages of learning, where both design and control steps are short, it becomes more pronounced as training advances (the execution steps become much longer). When employing randomly sampled experiences for training, the design policy, given the separate policy representation, tends to receive fewer updates compared to the control policy. Consequently, it can quickly become optimized only for a limited region around the local morphological optimum, which hinders the effective exploration of the design space, as shown in Section 5.2.

To address this concern, our co-design policy (actor network) is designed to facilitate more information sharing (Figure 2). While directly representing the intricate interplay between morphology and controller is challenging, we employ shared parameters to implicitly capture their relationships. This approach guarantees that both the "brain" and the "body" of a robot are derived from the same set of parameters and developed together.

**Unified State-Action Space** Our study focuses on co-designing flexible MSRs comprising various types of blocks, also known as voxels. Each voxel in the design space is represented by a discrete value that corresponds to its material or type of actuator (e.g., empty voxel=0, soft voxel=1, rigid voxel=2, horizontal actuator=3 and vertical actuator=4). In practice, we employ one-hot encoding to represent these values. The co-design policy is uniformly denoted as $\pi_\theta(a_t|s_t)$ and integrated into the aforementioned design-and-control MDP. Here, $s_t = \{s_t^d, s_t^c\}$ represents the concatenation of the design observation $s_t^d$ and control observation $s_t^c$ at time step $t$ in each episode. During the design stage, $s_t^c$ of $s_t$ will be set to zero and during the control stage, $s_t^d$ of $s_t$ will be consistent with the state of the last design step and unchanged. Precisely, with $N$ denoting the size of the design space (e.g., $N = 25$ for a $5 \times 5$ design space), we define $s_t^d = \{s_t^{d_1}, s_t^{d_2}, ..., s_t^{d_N}\}$, where $s_t^{d_i}$ for voxel $i$ is a vector comprising its type and the types of its Moore neighborhood, as shown in Appendix A. $s_t^c$ is the control observation, which comprises local observations from all voxels, denoted as $s_t^c = \{s_t^v, s_t^g\}$. $s_t^v = \{s_t^{v_1}, s_t^{v_2}, ..., s_t^{v_N}\}$ represents modular observations, where $s_t^{v_i}$ includes the relative position of each voxel's four corners with respect to the center of mass of the robot. $s_t^g$ represents task-related observations, such as the terrain information.

We utilize two feed-forward neural networks to decode shared information from the transformer-based encoder. The output layer dimension of the design action decoder matches the total number of material types (5 in our work), while the output layer dimension of the control action decoder is set to 1. As we model the co-design of modular soft robots as a sequence-to-sequence task, both the design and control actions have a length of $N$. During training, voxels are determined by sampling from a categorical distribution, which is formulated based on the output logits. During evaluation, the action corresponding to the highest logit value is selected. Additionally, the control action decoder generates the mean value $\mu$. By combining it with a constant standard deviation $\Sigma$, control signals can be sampled from this Gaussian distribution and then clipped within the range of $[0.6, 1.6]$, which corresponds to gradual contractions/expansions of the actuators. We use action masks to inform the policy whether an element in the output sequence is an actuator.

**Learning Process** Based on the unified MDP and standard RL practices, we use distributed trajectory sampling with multiple CPU threads to collect training data. Given that we have $K$ pre-training tasks (or environments in RL terminology), each task is allocated to its respective CPU thread.

Therefore, $K$ also signifies the number of threads we deploy. During each RL interaction, the state fed to our policy is represented as $\{s_t^{task_1}, s_t^{task_2}, ..., s_t^{task_K}\}$, which can be viewed as a uniform sampling process. Note that the time step $t$ here may vary among tasks. We train our co-design policy using PPO [58], which is based on the popular actor-critic architecture. The critic network shares the same architecture as the actor network (Figure 2), and it computes the value function, indicating a probable policy distribution. Its output is a $N \times 1$ continuous-valued vector where each element corresponds to the estimated value of a voxel. Here, we represent the overall morphology value by averaging the values across all voxels. With the policy gradient technique, the co-design policy is updated to optimize the predicted morphology value. After completing the pre-training phase, the resulting pre-trained co-designer is utilized to generate initial designs and control policies, which are subsequently fine-tuned using PPO again for unseen tasks.

## 5   Experiments

In this section, we experimentally evaluate our proposed approach to answer the following questions: (1) Does our method, PreCo, effectively perform brain-body pre-training and discover robot morphologies capable of executing multiple tasks? (2) How well does our method demonstrate zero-shot generalization and brain-body fine-tuning capabilities when faced with unseen co-design tasks? (3) What is the impact of the unified policy representations on the performance of PreCo?

### 5.1   Environments and Implementation

Based on the Evolution Gym platform [3], we establish a modular robot state-action space[2] that supports brain-body pre-training as described in Section 4. Our focus lies on a fixed design space of size $5 \times 5$ which includes 9 locomotion co-design tasks with open-ended environments: **Walker-v0 (easy)**, **PlatformJumper-v0 (hard)**, **UpStepper-v0 (medium)**, **ObstacleTraverser-v0 (medium)**, **BridgeWalker-v0 (easy)**, **GapJumper-v0 (hard)**, **DownStepper-v0 (easy)**, **ObstacleTraverser-v1 (hard)** and **Hurdler-v0 (hard)**, as shown in Figure 1. The difficulty levels are determined based on the performance of evolution-based co-design algorithms from the platform. For more information regarding the environmental details, please refer to Appendix B.

We work on the assumption that there are structural similarities between the pre-training and target co-design tasks. Guided by this understanding, we select the first four tasks for pre-training. The target tasks in our study essentially encompass different adaptations of the pre-training tasks, including the following types: (1) **More challenging scenarios**, such as the transition from ObstacleTraverser-v0 to ObstacleTraverser-v1 where the terrain becomes increasingly uneven; (2) **Transfers of comparable difficulty**, as seen when moving from Walker-v0 to BridgeWalker-v0 (with a shift to softer terrain) or from PlatformJumper-v0 to GapJumper-v0, wherein the gap between steps expands but the height of these steps reduces; (3) **"Reverse" scenarios**, exemplified by the transition from UpStepper-v0 to DownStepper-v0, where the direction of the steps is inverted.

We compare PreCo against the following baselines that are also suitable for multiple co-design tasks: (1) **PreCo-Sep**, which is based on PreCo but utilizes separate transformer-based design and control policies. This baseline allows us to investigate the effectiveness of the unified policy representation; (2) **CuCo** [5], a curriculum-based co-design method that consists of separate NCA-based design policy and transformer-based control policy. We set the curriculum of CuCo to be $3 \times 3 \rightarrow 5 \times 5$; (3) **MeCo**, which adopts the same network architecture as PreCo but is trained using Reptile [65], a popular meta-learning method. We use a 3-layer transformer encoder and run all experiments with the same number of policy iterations. More implementation details can be found in Appendix C.

### 5.2   Results

**Brain-Body Pre-Training**   We show the learning curves and converged robot morphologies of all methods in Figure 3. For each method, the learning curve is reported over 7 different runs. The

---

[2]https://github.com/Yuxing-Wang-THU/ModularEvoGym

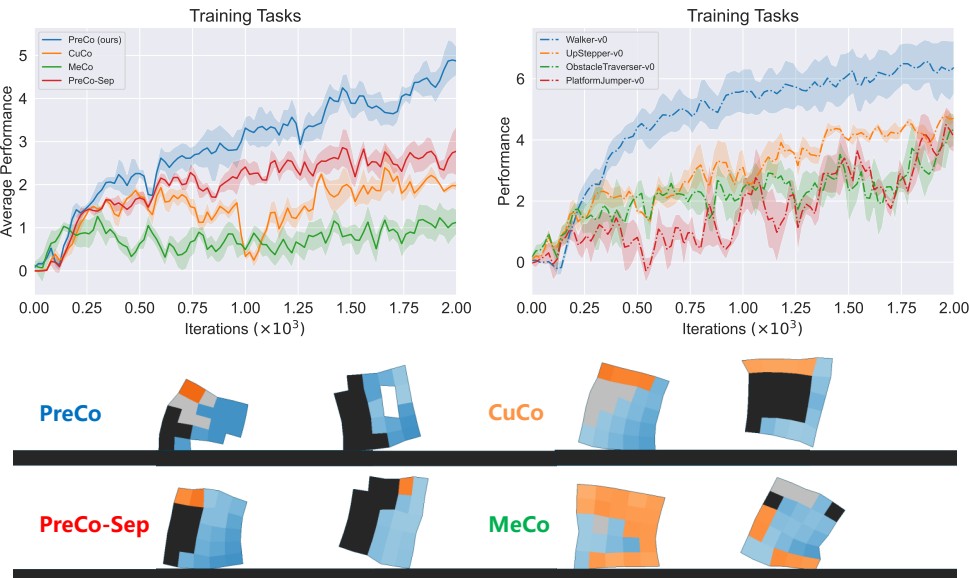

Figure 3: Learning curves and converged morphologies of brain-body pre-training. In the left figure, we demonstrate the mean and standard deviation of average task performance against the number of policy iterations for all methods. The right figure displays the individual learning curves of PreCo. The bottom figure shows two representative converged morphologies from each method.

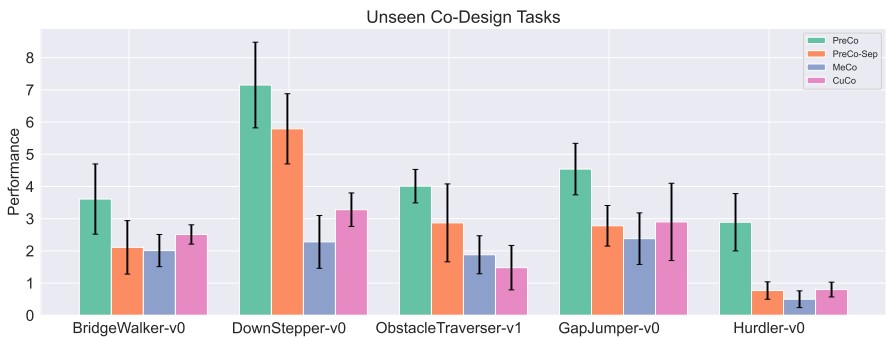

Figure 4: Evaluation of zero-shot generalization. The height of each bar represents the average zero-shot performance of a method, and error bars indicate the corresponding standard deviation.

figure clearly demonstrates that our proposed method, PreCo, outperforms the baselines in terms of learning speed and final performance. A comparison of the robot morphologies designed by each method reveals intriguing distinctions. Both PreCo and PreCo-Sep are capable of discovering robot bodies with rigid legs, which are essential for maintaining balance in various locomotion tasks. PreCo, in particular, exhibits the ability to utilize empty voxels, resulting in robots with serrated and hollow structures that potentially contribute to enhanced performance. On the other hand, CuCo produces robots with several repeated body segments. Although it benefits from curriculum learning, as evident from its learning curve, it converges to a local morphological optimum with lower performance. As for MeCo, the "meta-bodies" it discovered seem to prefer horizontal actuators, which may not be efficient in certain jumping tasks. However, our primary interest lies in evaluating its performance on the unseen co-design tasks.

**Zero-Shot Generalization** One of our objectives in this work is to train a co-design policy capable of generating a single modular soft robot that can generalize to unseen co-design tasks. Figure 4 illustrates that PreCo consistently achieves higher zero-shot performance across new environments

Table 1: Final performance across environments and baselines, each result is reported over 7 different runs. Methods on the left side of the divider are fine-tuned from their corresponding pre-trained models, while those on the right side are trained from scratch on target tasks.

| Environment | PreCo-FT | PreCo-Sep-FT | MeCo-FT | CuCo-FT | PreCo-Scratch | GA |
|---|---|---|---|---|---|---|
| BridgeWalker-v0 | $3.69 \pm 0.63$ | $3.11 \pm 0.91$ | $4.75 \pm 0.62$ | $4.37 \pm 0.61$ | $4.81 \pm 0.21$ | $\mathbf{5.31 \pm 0.25}$ |
| Hurdler-v0 | $\mathbf{4.10 \pm 0.52}$ | $1.73 \pm 2.19$ | $1.88 \pm 1.30$ | $1.66 \pm 1.24$ | $2.76 \pm 1.39$ | $1.51 \pm 0.25$ |
| DownStepper-v0 | $\mathbf{9.01 \pm 0.02}$ | $8.90 \pm 0.01$ | $4.58 \pm 0.74$ | $8.65 \pm 0.33$ | $8.98 \pm 0.01$ | $8.91 \pm 0.01$ |
| GapJumper-v0 | $\mathbf{5.78 \pm 1.25}$ | $2.58 \pm 0.19$ | $2.83 \pm 0.50$ | $3.58 \pm 0.87$ | $3.48 \pm 0.66$ | $3.35 \pm 0.15$ |
| ObstacleTraverser-v1 | $\mathbf{4.88 \pm 0.12}$ | $3.88 \pm 0.69$ | $3.03 \pm 0.50$ | $2.00 \pm 0.77$ | $3.38 \pm 0.69$ | $2.98 \pm 0.63$ |

when compared to the baseline methods. Figure 7 in Appendix D shows that the robot designed by PreCo demonstrates the ability to employ the skill of somersaulting for traversing challenging terrains, relying on its environmental comprehension. Furthermore, it exhibits an understanding of the necessity to lean back to preserve stability while descending stairs.

**Brain-Body Fine-Tuning** Besides the evaluation of zero-shot generalization, we also consider a more general setting that allows the co-design policy to fine-tune its parameters to adapt to target tasks. We aim to investigate whether brain-body fine-tuning is better than training from scratch. Keeping this in mind, we introduce PreCo-Scratch and GA [3] as additional baselines, all of which are trained from scratch on target tasks. Here, we limit the number of brain-body fine-tuning iterations to 300 and policy iterations of learning from scratch to 2000. Table 1 presents all results across unseen environments. It is evident that PreCo outperforms the baseline algorithms in most environments. For morphological results, Figure 8 in Appendix D demonstrates that PreCo exhibits intelligent behavior by retaining the beneficial serrated structure for effective stair climbing and obstacle traversal, while also making adaptive modifications to suit the new environment.

**The Shared Policy Representation** To go a step further, in Appendix E, we provide a performance comparison between PreCo and PreCo-Sep when trained from scratch across 10 co-design tasks (Table 3). Figure 9 and Figure 10 illustrate their learning processes in a complex task, Climber-v0, which requires the policy to have a good exploration ability to grow irregular structures. Clearly, PreCo exhibits the ability to explore beyond the local morphological optimum, allowing it to develop thin limbs that aid in climbing. In summary, the shared policy representation creates additional opportunities for exploring the design space. This is because, as the parameters of the "control policy" undergo adjustments, the "design policy" is concurrently updated.

## 6 Limitations and Conclusion

We have introduced PreCo, a co-design method that utilizes brain-body pre-training to generate modular soft robots capable of performing multiple tasks. Through the adoption of shared policy representations, which capture the inherent brain-body connections across various co-design tasks, we have observed its favorable zero-shot generalization and few-shot adaptation capabilities in addressing previously unseen co-design tasks.

There are a number of areas for improvement. As shown in Table 1, our method does not perform very well in BridgeWalker. This might be because the selection of training tasks does not cover the variation of soft terrain, potentially leading to ambiguities in the co-design policy. Exploring the selection of pre-training co-design tasks could be interesting for future research. Additionally, although our policy representation appears to facilitate the learning process, it is worth noting that destructive mutations of the network parameters can still occur. Further investigation into the genotype-phenotype-fitness mapping of this policy would be valuable. In our paper, we tested our method using a simulator with relatively fundamental modules as a proof of concept to show its effectiveness, developing a general pipeline for translating learned models to physical will definitely be our next step. We also provide a detailed discussion of this sim-to-real issue in Appendix F and envision our method serving as a foundation for subsequent research which tends to make the co-design of modular robots more practical both in the simulation and the real world.

**Acknowledgments**

We sincerely thank the anonymous reviewers for their helpful comments in revising the paper. This work was supported by the National Key R&D Program of China (2022YFB4701400/4701402).

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

# A    Parameterization of the Design Space

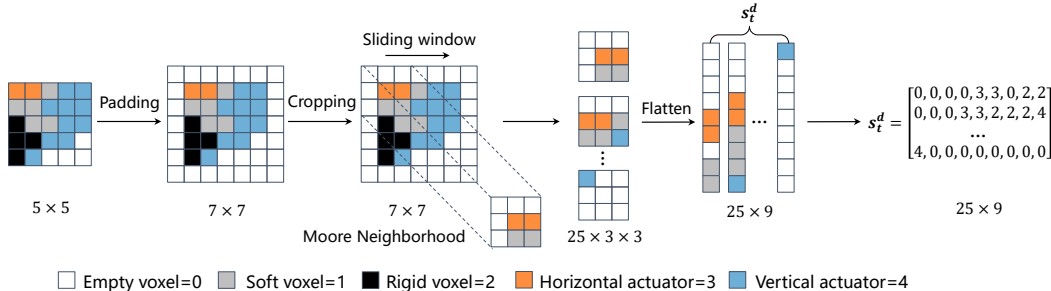

Figure 5: Parameterization of the design space. Initially, the design space is surrounded by empty voxels. Each voxel is denoted by a discrete value, reflecting its material characteristic. Then, a sliding window is used to get each voxel's local state, which is composed of its type and the types of its Moore neighbors. Finally, the design state is formulated as an ordered sequence.

# B    Environment Details

Our experiments are based on the simulation platform from [3, 5]. In this section, we provide additional details of the used environments (Figure 6).

**Position.** $p^o$ is a 2-dim vector that represents the position of the center of mass of an object $o$ in the simulation at time $t$. $p_x^o$ and $p_y^o$ are $x$ and $y$ components of this vector, respectively. $p^o$ is calculated by averaging the positions of all the point-masses that make up object $o$ at time $t$.

**Velocity.** $v^o$ is a 2-dim vector that represents the velocity of the center of mass of an object $o$ in the simulation at time $t$. $v_x^o$ and $v_y^o$ are $x$ and $y$ components of this vector, respectively. $v^o$ is calculated by averaging the velocities of all the point-masses that make up object $o$ at time $t$.

**Orientation.** $\theta^o$ is a 1-dim vector that represents the orientation of an object $o$ in the simulation at time $t$. Let $p_i$ be the position of point mass $i$ of object $o$. $\theta^o$ is computed by averaging over all $i$ the angle between the vector $p_i - p^o$ at time $t$ and time 0. This average is a weighted average weighted by $||p_i - p^o||$ at time 0.

**Other observations.** $h_b^o(d)$ is a vector of length $(2d+1)$ that describes elevation information around the robot below its center of mass. More specifically, for some integer $x \leq d$, the corresponding entry in vector $h_b^o(d)$ will be the highest point of the terrain which is less than $p_y^o$ between a range of $[x, x+1]$ voxels from $p_x^o$ in the $x$-direction.

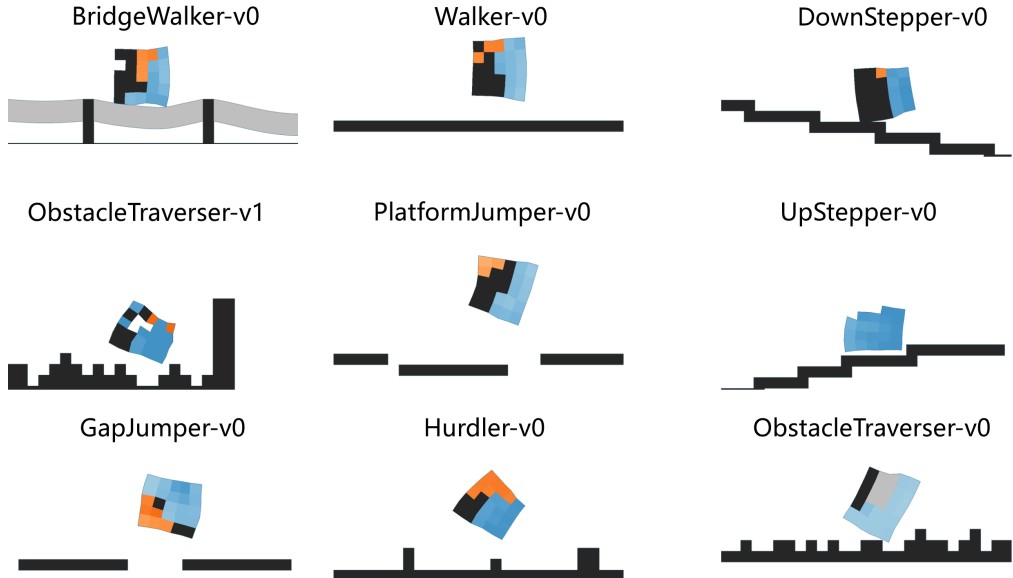

Figure 6: Visualization of all environments used in our work.

## B.1 Walker-v0

In this task, the robot is rewarded by walking as far as possible on flat terrain. The task-specific observation is $v^{robot}$, and the reward $R$ is:

$$R = \Delta p_x^{robot} \tag{2}$$

which rewards the robot for moving in the positive $x$-direction. The robot receives a reward of 1 for reaching the end of the terrain. The episode duration reaches a 500 time steps.

## B.2 BridgWalker-v0

In this task, the robot is rewarded by walking as far as possible on a soft rope-bridge. The task-specific observation is $\{v^{robot}, \theta^{robot}\}$, and the reward $R$ is:

$$R = \Delta p_x^{robot} \tag{3}$$

which rewards the robot for moving in the positive $x$-direction. The robot receives a reward of 1 for reaching the end of the terrain. The episode duration reaches a 500 time steps.

## B.3 Upstepper-v0

In this task, the robot climbs up stairs of varying lengths. The task-specific observation is formed by concatenating vectors $\{v^{robot}, \theta^{robot}, h_b^{robot}(5)\}$, and the reward $R$ is:

$$R = \Delta p_x^{robot} \tag{4}$$

which rewards the robot for moving in the positive $x$-direction. The robot also receives a one-time reward of 2 for reaching the end of the terrain. The episode duration reaches a 600 time steps.

## B.4 Downstepper-v0

In this task, the robot climbs down stairs of varying lengths. The task-specific observation is formed by concatenating vectors $\{v^{robot}, \theta^{robot}, h_b^{robot}(5)\}$, and the reward $R$ is:

$$R = \Delta p_x^{robot} \tag{5}$$

which rewards the robot for moving in the positive $x$-direction. The robot also receives a one-time reward of 2 for reaching the end of the terrain, and a one-time penalty of $-3$ for rotating more than 90 degrees from its originally orientation in either direction. The episode duration reaches a 500 time steps.

### B.5   ObstacleTraverser-v0

In this task, the robot walks across terrain that gets increasingly more bumpy. The task-specific observation is formed by concatenating vectors $\{v^{robot}, \theta^{robot}, h_b^{robot}(5)\}$, and the reward $R$ is:

$$R = \Delta p_x^{robot} \tag{6}$$

which rewards the robot for moving in the positive $x$-direction. The robot also receives a one-time reward of 2 for reaching the end of the terrain, and a one-time penalty of $-3$ for rotating more than 90 degrees from its originally orientation in either direction. The episode duration reaches a 1000 time steps.

### B.6   ObstacleTraverser-v1

In this task, the robot walks through very bumpy terrain. The task-specific observation is formed by concatenating vectors $\{v^{robot}, \theta^{robot}, h_b^{robot}(5)\}$, and the reward $R$ is:

$$R = \Delta p_x^{robot} \tag{7}$$

which rewards the robot for moving in the positive $x$-direction. The robot also receives a one-time reward of 2 for reaching the end of the terrain. The episode duration reaches a 1000 time steps.

### B.7   Hurdler-v0

In this task, the robot walks across terrain with tall obstacles. The task-specific observation is formed by concatenating vectors $\{v^{robot}, \theta^{robot}, h_b^{robot}(5)\}$, and the reward $R$ is:

$$R = \Delta p_x^{robot} \tag{8}$$

which rewards the robot for moving in the positive $x$-direction. The robot also receives a one-time penalty of $-3$ for rotating more than 90 degrees from its originally orientation in either direction. The episode duration reaches a 1000 time steps.

### B.8   GapJumper-v0

In this task, the robot traverses a series of spaced-out floating platforms all at the same height. The task-specific observation is formed by concatenating vectors $\{v^{robot}, \theta^{robot}, h_b^{robot}(5)\}$, and the reward $R$ is:

$$R = \Delta p_x^{robot} \tag{9}$$

which rewards the robot for moving in the positive $x$-direction. The robot also receives a one-time penalty of $-3$ for falling off the platforms. The episode duration reaches a 1000 time steps.

### B.9   PlatformJumper-v0

In this task, the robot traverses a series of floating platforms at different heights. The target design space is $5 \times 5$. The task-specific observation is formed by concatenating vectors $\{v^{robot}, \theta^{robot}, h_b^{robot}(5)\}$, and the reward $R$ is:

$$R = \Delta p_x^{robot} \tag{10}$$

which rewards the robot for moving in the positive $x$-direction, The robot also receives a one-time penalty of $-3$ for rotating more than 90 degrees from its originally orientation in either direction or for falling off the platforms (after which the environment resets). The episode duration reaches a 1000 time steps.

## C  Implementation Details

### C.1  Hyperparameters and Training Procedure

We use PyTorch [66] to implement all the models used in our work. We take the official implementation of transformer from Pytorch which uses ***TransformerEncoderLayer*** module, and add a learnable position embedding. All hyperparameters of PreCo are listed in Table 2.

Our co-design policy can be trained in an end-to-end RL manner because we unify the design and control processes as a single MDP. That is, at the start of each RL episode, the policy first takes a finite number of design actions to develop a robot morphology, and no reward is assigned to the policy during this period. Subsequently, the resulting robot is controlled by this policy to collect the environmental rewards, which also provides learning signals for the design actions. Once the desired number of trajectories is collected using distributed trajectory sampling (described in Section 4), the policy is updated using PPO. We also ensure that the baselines and our method use the same number of policy iterations (simulation steps) for optimization.

Table 2: Hyperparameters of PreCo.

|  | Hyperparameter | Value |
|---|---|---|
| | GAE | True |
| | GAE $\lambda$ | 0.95 |
| | Learning rate | $2.5 \cdot 10^{-4}$ |
| | Linear learning rate decay | True |
| | Clip parameter | 0.1 |
| | Value loss coefficient | 0.5 |
| | Entropy coefficient | 0.01 |
| | Time steps per rollout | 5120 |
| PPO | Optimizer | Adam |
| | Evaluation interval | 10 |
| | Discount factor $\gamma$ | 0.99 |
| | Clipped value function | True |
| | Observation normalization | True |
| | Observation clipping | $[-10, 10]$ |
| | Reward normalization | True |
| | Reward clipping | $[-10, 10]$ |
| | Policy epochs | 8 |
| | Neighborhood | Moore |
| | Design steps | 1 |
| | Number of layers | 3 |
| | Number of attention heads | 1 |
| Transformer | Embedding dimension | 128 |
| | Feedforward dimension | 256 |
| | Non linearity function | ReLU |
| | Dropout | 0.0 |

### C.2  Details of the Baseline Algorithms

For baseline algorithms, we use the official implementation of GA from Evolution Gym [3] and employ a population of 12 agents. It's worth noting that the inner loop of control optimization is also driven by PPO, while the outer loop of morphology optimization is implemented using the evolutionary algorithm. Additionally, we use the official implementation of CuCo from [5] and Reptile from [65]. In the remaining section, we demonstrate details about these baselines.

**GA** GA directly encodes the robot's morphology as a vector where each element is tailored to the voxel's material property in order. It uses elitism selection and a simple mutation strategy to evolve the population of robot designs. The selection keeps the top $x\%$ of the robots from the current population as survivors and discards the rest, and the mutation can randomly change each voxel of the robot with a certain probability (mutation rate). In our study, the survivor rate starts at $60\%$ and decreases linearly to $0\%$, and the mutation rate is set to $10\%$.

**CuCo** CuCo is a curriculum-based co-design method that consists of separate NCA-based design policy and transformer-based control policy. This curriculum-based method expands the design space from a small size to the target size using reinforcement learning with a predefined curriculum. In our study, we set the curriculum of CuCo to be $3 \times 3 \rightarrow 5 \times 5$ and adhere to the original hyperparameter settings of CuCo as presented in [5].

**MeCo** MeCo utilizes the same network architecture as PreCo but is trained with the Reptile [65], a popular meta-learning method. Reptile is designed to identify model parameters that serve as an optimal starting point for adaptation across various tasks. When encountering a novel task, the model is expected to need fewer updates or episodes to achieve proficient performance. In contrast to another meta-learning method, MAML[67], which necessitates second-order gradients (gradients of gradients) during its meta-update step, Reptile simply averages the updates. This characteristic makes Reptile more computationally efficient and easier to implement. In our study, we set the meta-learning rate to $0.25$, and the update iteration for each training task is configured to be $20$.

### C.3 Computational Cost

We use distributed trajectory sampling with multiple CPU threads to collect training data (described in Section 4). For pre-training experiments in the paper, it takes around 2 days to train our model on a standard server with 40 CPU cores and an NVIDIA RTX 3090 GPU.

## D Visualization Results

In this section, we provide some visualization results of brain-body pre-training and brain-body fine-tuning, as shown in Figure 7 and Figure 8, respectively.

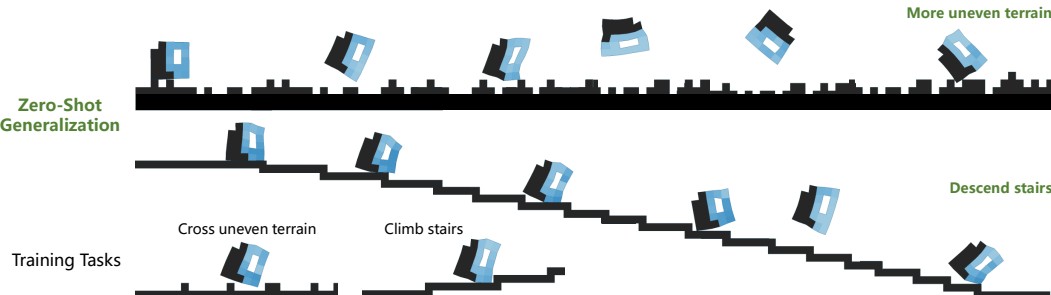

Figure 7: Visualization of PreCo's zero-shot behavior. The figure shows screenshots at consecutive time intervals of the designed robot's behavior. Compared with training tasks, We find that PreCo shows favorable generalization properties and intriguing behavior when facing new environments (beginning from $00 : 28$ in this video).

## E Ablation of the Shared Policy Representation

The performance results of PreCo and PreCo-Sep in Figure 3 and Table 1 suggest that a shared policy representation facilitates zero-shot generalization and few-shot adaptation, surpassing methods that employ separated representations. Furthermore, we present a comparison of the performance of

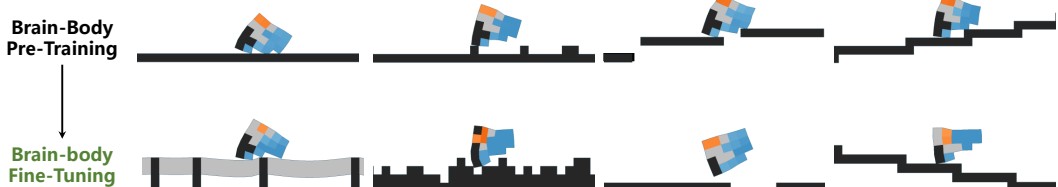

Figure 8: Visualization of brain-body fine-tuning. The pre-trained co-design policy shows the ability to swiftly adjust both the morphology and control strategy to adapt to new co-design tasks (beginning from $00 : 42$ in this video).

Table 3: Performance results across 10 co-design tasks. All methods are trained from scratch.

| Environment | PreCo-Scratch | PreCo-Sep-Scratch |
|---|---|---|
| Walker-v0 | $10.47 \pm 0.01$ | $10.46 \pm 0.01$ |
| Climber-v0 | $\mathbf{2.23 \pm 0.87}$ | $0.43 \pm 0.02$ |
| Hurdler-v0 | $\mathbf{2.76 \pm 1.39}$ | $2.07 \pm 1.33$ |
| UpStepper-v0 | $\mathbf{7.23 \pm 1.46}$ | $4.06 \pm 0.28$ |
| DownStepper-v0 | $\mathbf{8.98 \pm 0.01}$ | $7.46 \pm 0.71$ |
| GapJumper-v0 | $3.48 \pm 0.66$ | $\mathbf{3.51 \pm 0.53}$ |
| BridgeWalker-v0 | $4.81 \pm 0.21$ | $\mathbf{5.46 \pm 1.01}$ |
| PlatformJumper-v0 | $\mathbf{6.12 \pm 0.82}$ | $3.97 \pm 0.33$ |
| ObstacleTraverser-v0 | $\mathbf{6.03 \pm 2.34}$ | $5.08 \pm 0.19$ |
| ObstacleTraverser-v1 | $\mathbf{3.38 \pm 0.69}$ | $2.78 \pm 1.06$ |

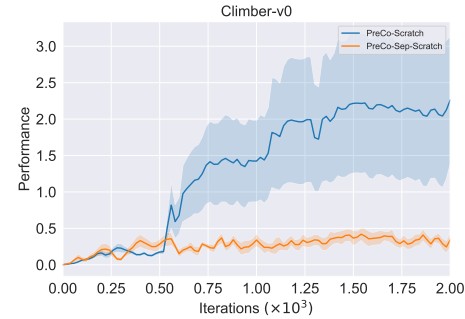

Figure 9: Learning curves in Climber-v0.

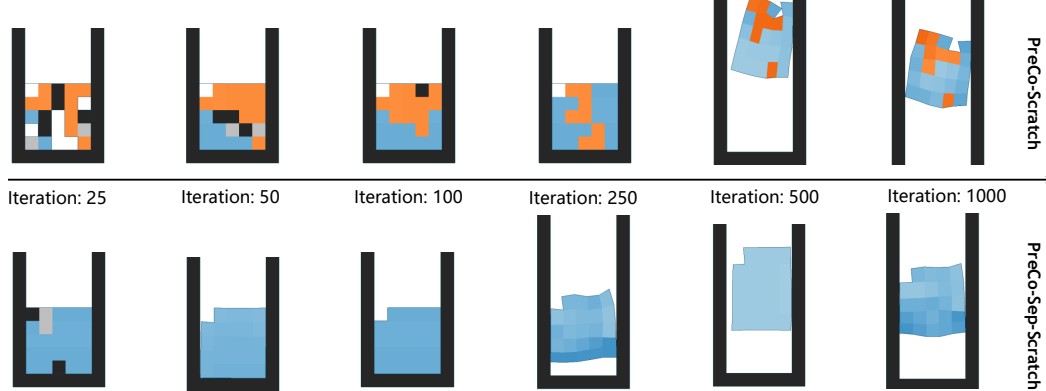

Figure 10: Morphological results of the two methods obtained during their learning processes.

these two methods when trained from scratch across 10 co-design tasks, the results are shown in Table 3. Figure 9 and Figure 10 illustrate their learning processes in a complex task, Climber-v0.

## F  Discussion of the Sim-to-Real Issue

In our paper, we tested our method using a simulator with relatively fundamental modules as a proof of concept to show its effectiveness. In this section, We discuss the sim-to-real issue of our work.

From the perspective of "Sim", the EvolutionGym platform [3] used in our work employs several simplifications to reach a trade-off between the simulation quality and velocity. Thus, for more realizable sim-to-real transfer, an improved version of its physics engine is needed to model soft-body physics in 3D space. We believe that using the Finite Element Method (FEM) numerical simulation would be one of the feasible ways to narrow this sim-to-real gap because the voxel-based design provides a naturally organized mesh, and its resolution and element type could be relatively

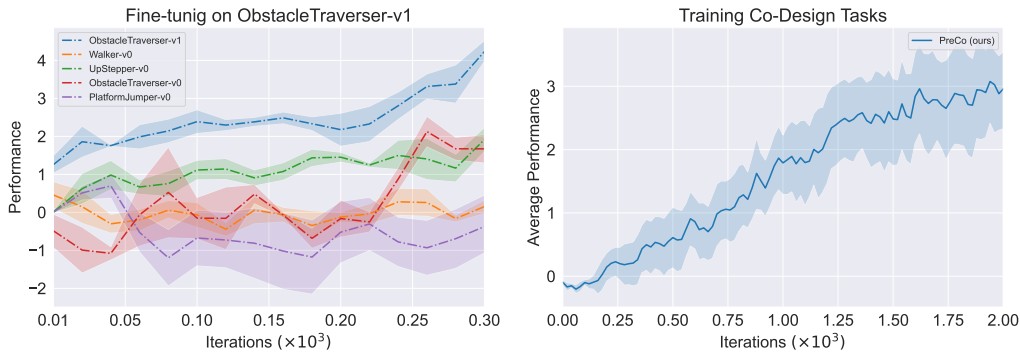

Figure 11: Additional experiments. Left: visualization of the fine-tuning process. Right: learning curve of brain-body pre-training on more diverse co-design tasks.

easily determined. Moreover, FEM has shown promise for real-world soft robotic models when combined with highly stable implicit Euler integration [42].

In our study, we focus on the model-free co-design problem where system modeling is not required, and we use reinforcement learning to approximate the gradient of design and control. This endeavor requires a large amount of training data. However, when considering the real-world problem, it is possible to transfer to the model-based setting. By leveraging the differentiable properties of certain FEM models [40], the universal parameterized co-design policy can be updated using Back-Propagation Through Time (BPTT), resulting in more efficient brain-body pre-training. When we move to the 3D modular robot design space, we would like to adapt the parameterization method depicted in Figure 5 to its 3D version. By using a "3D convolutional kernel", we can easily create input design observation sequences for the transformer-based policy. Although transformer has sufficient capacity to handle long sequences, a more expansive design space (e.g., thousands of voxels) would necessitate a more complex transformer model. This, in turn, could demand significantly higher computational power.

From the perspective of "Real", one of the foremost considerations is the material selection, which should be predicated upon the required flexibility, durability, and functionality [68, 69]. With this criterion, the Diels-Alder (DA) polymer [70] or silicone voxels [38], in conjunction with multi-material cubic blocks produced through 3D printing, may serve as ideal components for constructing the body of a MSR. To establish the local observation space, each voxel could be equipped with an array of sensors, such as touch, pressure and velocity sensors. Alternatively, soft sensors, crafted from conductive elastomers that alter resistance upon deformation, could offer valuable feedback to the control system. Furthermore, Peano-HASEL actuators [71] or pneumatic actuators [72] might be suitable for volumetric actuation (probably limited to expansion for efficient simulation), and closed-loop control could be achieved by utilizing Neural Networks (NNs). In the real-world setting, factors like material imperfections, air resistance, friction and many others come into play, we also need to iteratively refine the design and control algorithms based on real-world feedback.

We acknowledge that each point discussed above presents its challenges but is well worth in-depth investigation, and we aspire for our work to serve as a catalyst for future research into the co-design of modular soft robots.

## G  Additional Experiments

### G.1  How Does Fine-Tuning on the Target Task Affect Performance on the Training Tasks?

It is worth noting that when a pre-trained co-design policy undergoes fine-tuning for a new target task, it can experience what's known as "catastrophic forgetting". This means that it might for-

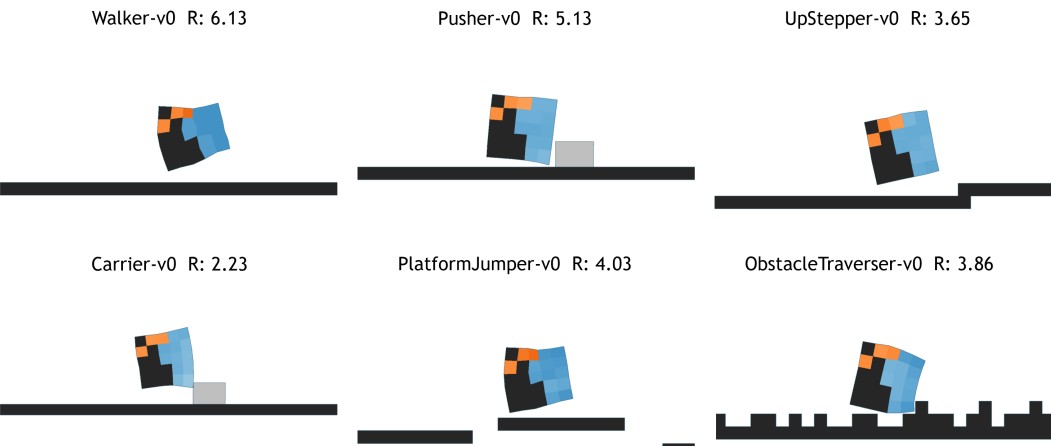

Figure 12: Morphological results of brain-body pre-training.

get certain information or patterns it learned during the pre-training phase, potentially leading to a decrease in performance on the original training tasks.

We track the performance changes on 4 pre-training tasks when the co-design policy is fine-tuned on ObstacleTraverser-v1. The left side of Figure 11 illustrates a consistent decrease in performance for Walker-v0 and PlatformJumper-v0 due to the significant disparity between the target task and the original tasks. In contrast, If the target task is similar to the pre-training tasks, the co-design policy might retain more of its initial knowledge. For instance, performance on tasks like UpStepper-v0 and ObstacleTraverser-v0 remains less affected throughout the fine-tuning.

## G.2 Pre-Training on More Diverse Co-design Tasks

In our paper, we select 4 locomotion tasks for pre-training and 5 tasks for testing. As we focus on co-designing modular soft robots to perform multiple tasks, the wealth of brain-body links embedded within the enormous combined search space offers sufficient diversity for effective policy learning.

To further explore the potential of PreCo, we also conduct an additional experiment that encompasses more diverse co-design tasks for pre-training. In this experiment, except for the original 4 pre-training tasks, we add Pusher-v0 (the robot is encouraged to push a box initialized in front of it as far as possible) and Carrier-v0 (the robot is encouraged to carry a box initialized above it as far as possible) to the co-design policy's learning procedure. Moreover, we add Thrower-v0 (the robot is encouraged to throw a box initialized above it as far as possible) to the test tasks. The right side of Figure 11 demonstrates the learning curves, and the result is averaged over 5 different runs. The results of Figure 12, Figure 13 and Figure 14 show that PreCo still performs well on zero-shot generalization and few-shot adaptation.

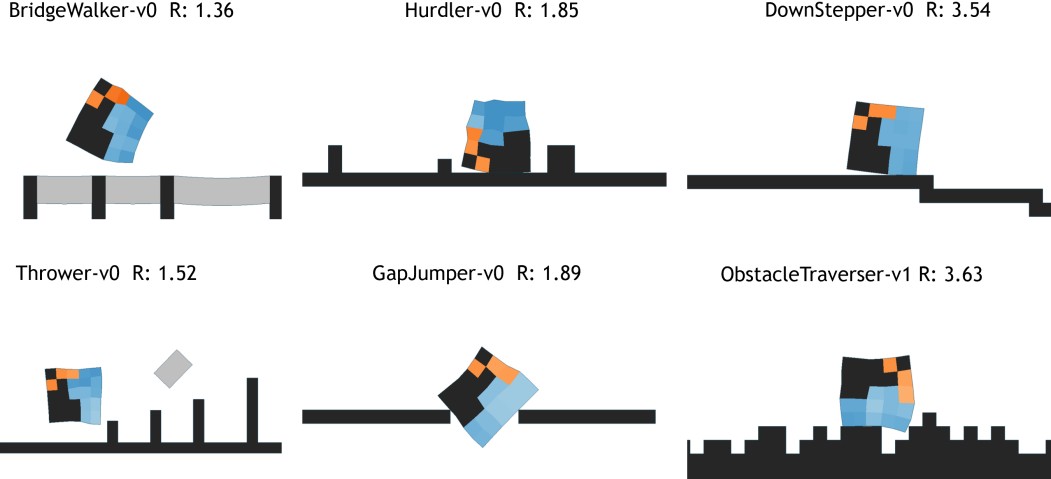

Figure 13: Morphological results of zero-shot generalization.

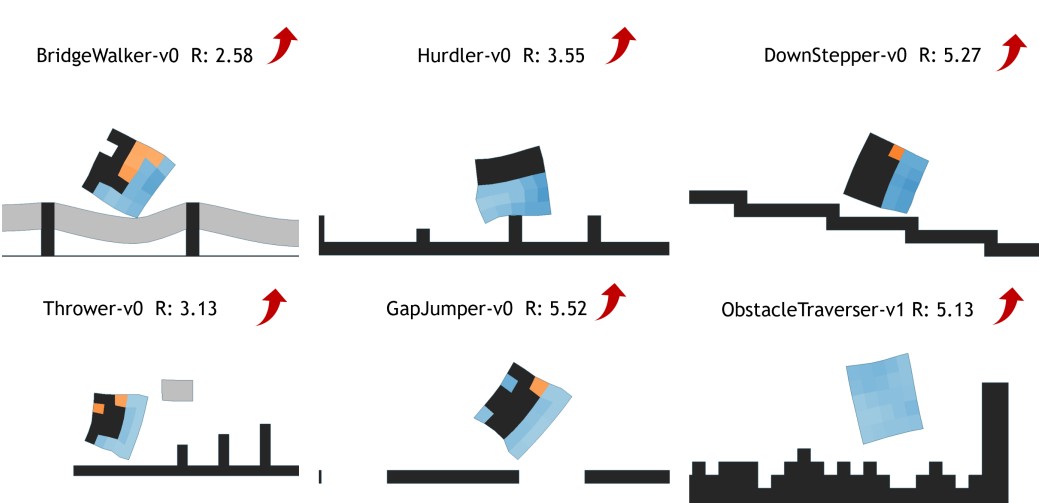

Figure 14: Morphological results of brain-body fine-tuning.

