# OpenReview forum: "PreCo: Enhancing Generalization in Co-Design of Modular Soft Robots via Brain-Body Pre-Training"
_robot-learning.org/CoRL/2023/Conference — CoRL 2023 Oral_

### Official Review · Reviewer_Bcvj · 2023-06-27

**Confidence:** 4
**Originality:** Very Good
**Technical Quality:** Very Good
**Clarity Of Presentation:** Very Good
**Impact:** 4

**Recommendation:**

Weak Accept: I recommend accepting the paper, but will not argue for my recommendation if the majority of other reviewers have a different opinion.

**Review:**

The introduction of the PreCo concept is novel, with practical implications for bolstering generalization within the co-design procedure for modular soft robots. The paper offers incisive empirical evidence concerning the efficacy of brain-body pre-training, zero-shot generalization, and fine-tuning. The structure of the pleiotropic co-design policy is robust and eliminates the requirement for a robot population, leading to enhanced flexibility and sample efficiency.

Also, a comprehensive and well-organized review of pertinent literature is presented. The experiments conducted on the modular soft robot system are thorough and meticulous, providing a robust foundation for the proposed methodology.

However, the experiments do not encompass a diverse range of tasks, and further investigation into the choice of co-design tasks could serve as an intriguing area for future research. The manuscript could benefit from a comparison or discussion with established baselines in the field, thereby providing a more informed understanding of the degree of improvement proffered by the proposed methodology. Details for replication of the study are limited, and a more exhaustive evaluation and ablation studies of the proposed method are required.

**Quality Of The Limitations Section:**

Limitations are addressed clearly

**Questions For Rebuttal:**

Could the authors offer more insight into the choice of co-design tasks and the rationale behind their selection?
Is it possible to compare the proposed methodology with recognized baselines in the field to offer a more comprehensive understanding of its level of improvement?
Would the authors be able to provide further details for the replication of the study, in addition to more extensive evaluation and ablation studies of the proposed method?
Is it possible to conduct physical experiments?


**Robotics Focus:**

Highly relevant to robotics but no hardware experiments

**Summary Of Paper:**

The paper presents a novel approach dubbed PreCo, which incorporates brain-body pre-training within the co-design framework for modular soft robots, aimed at bolstering their generalization and adaptability capabilities. The methodology hinges on the encoding of co-design principles into models achieved through pre-training a versatile co-design policy on a wide-ranging set of tasks. This pre-trained co-designer is subsequently employed to generate preliminary designs and control policies, which are then fine-tuned specific to a co-design task. The results reveal not only zero-shot generalization to unseen co-design tasks but also expedited adaptation with substantially fewer iterations. The work provides noteworthy contributions to co-design in robotics, more specifically within the field of modular soft robots.



**Summary Of Recommendation:**

The manuscript provides substantial contributions to the domain of co-design in robotics, particularly regarding modular soft robots, by presenting an innovative and pragmatic methodology, coupled with astute empirical findings and a comprehensive review of relevant literature. However, the study does fall short in certain aspects such as limited implementation details and hardware demo and evaluation studies, which necessitate further clarification.

---

### Official Review · Reviewer_dKrr · 2023-07-17

**Confidence:** 3
**Originality:** Good
**Technical Quality:** Very Good
**Clarity Of Presentation:** Good
**Impact:** 3

**Recommendation:**

Weak Accept: I recommend accepting the paper, but will not argue for my recommendation if the majority of other reviewers have a different opinion.

**Review:**

### Strengths

- The authors clearly motivate their use of a single backbone network for design and control, grounding it in a biological analogy.
- The proposed method yields impressive empirical results.

### Weaknesses

- The authors identify a number of challenging aspects of the co-design problem, but it is not clear how their approach solves all of these challenges. For example:
	- Challenge: since relatively few design actions occur early in each rollout (compared to far more control actions), there is relatively less reward information on which to train the design policy. The authors claim that a shared backbone helps with this, and their ablation `PreCo-Sep` supports this.
	- Challenge: different design morphologies result in action spaces of different dimension. The authors claim that a transformer architecture helps with this, but they don't explain how. Please add more detail here.
- You say that the genetic algorithm is used to provide an estimate of the upper bound performance for each task, but the GA doesn't seem to learn well on any task. Why is that?
- The limitations section is not very comprehensive. In addition to the two limitations discussed (poor performance on the bridge environment and the risk of destructive mutations/forgetting), I can think of several others:
	- How would you generalize to robots in three dimensions? Would this problem suffer from a curse of dimensionality (e.g. 3D convolutions tend to be substantially harder than 2D convolutions; would a transformer-based architecture share this problem)?
	- How would you fabricate a robot with one of these designs? How could you enforce manufacturability constraints? Are there existing robot architectures or manufacturing methods that are compatible with the types of designs that your method produces?

### Minor comments

- "Transformer" is commonly written lower-case as "transformer" unless at the start of a sentence.
- When discussing your results, please don't use terms like "remarkable" or "astonishing". It's up to the reader whether they are astonished or not.

**Quality Of The Limitations Section:**

Additional details required

**Questions For Rebuttal:**

Please address all of the questions and issues raised above. In addition:

1. How are the design and control actions represented?
2. Why do you train on a CPU? Wouldn't GPU training be more effective?
3. Page 5, line 177-8: "We estimate the overall value of the morphology by taking the average of the values assigned to each individual voxel." What does this mean? What is the "overall value of the morphology" and how is it used?
4. You claim that the single backbone ("pleiotropic representation") yields better preformance. Can you comment on why MeCo (which also uses this shared representation) fails to learn or generalize on most tasks? Why does your pretraining work while the Reptile pre-training used by MeCo fail on most tasks?
5. What is CPPN-NEAT, and how does it compare to your method?

**Robotics Focus:**

Relevant but unlikely to deploy to hardware in near future

**Summary Of Paper:**

This paper studies the problem of automatically synthesizing designs and controllers for soft robots that are capable of generalizing (either zero-shot or with some fine tuning) to new tasks. Specifically, they study the co-design problem for 2D soft robots in environments of varying difficulties. The main innovation in the paper appears to be the use of a single shared backbone network (a transformer) for both the design and control policy networks (rather than separate representations as in prior works). The authors demonstrate better zero-shot and fine-tuned transfer to previously unseen tasks on a range of problems.

**Summary Of Recommendation:**

This paper presents a simple idea (a shared design/control policy backbone plus pre-training) that yields good empirical results in simulation. I think the core idea is worth sharing, but I recommend that the authors invest a considerable amount of time in revising and making sure that their explanation is clear. Please take care to disentangle causes and effects in your explanations; there are a number of distinct challenges in co-design, and you propose a number of distinct solutions, so it is important that you help the reader understand which solutions help address which challenges (both through explanation and empirical ablation).

**Update post-rebuttal** I have no further questions and am content to leave my score at weak accept.

---

### Official Review · Reviewer_WFDJ · 2023-07-19

**Confidence:** 4
**Originality:** Very Good
**Technical Quality:** Very Good
**Clarity Of Presentation:** Excellent
**Impact:** 3

**Recommendation:**

Strong Accept: I recommend accepting the paper and will argue for my recommendation even if other reviewers hold a different opinion.

**Review:**

The paper is well organized and the writing is clear. The presented methods are sound. The context of this paper is interesting because traditionally co-design methods focus on optimizing for "one robot one task", unlike biological system, while this paper presents a new learning approach aiming to address multiple co-design tasks simultaneously. The authors used modular soft robots to build their case allowing them to construct robot bodies by combining various types of deformable cubes and adjusting their volume to generate control signals. Aiming to design a single modular soft robot that achieves a wide variety of tasks, the paper first train the robot by adapting multi-task reinforcement learning. Then, the robot is further trained on unseen tasks. The authors implemented a new co-design policy, namely pleiotropic co-design policy, allowing them to integrate design and control into a unified decision process.

The authors illustrate that their proposed method could be more effective in learning and executing multiple tasks against various baseline models in simulation environment. However, the morphologies that their approach generated for locomotion purpose on various terrain profiles seem unconventional and not relevant to legged or wheeled locomotion morphologies. It would be interesting to see that their method converge to an efficient morphology for locomotion. Maybe varying the reward function towards legged or wheeled appendages might serve such a purpose.

**Quality Of The Limitations Section:**

Limitations are addressed clearly

**Questions For Rebuttal:**

Provide a reference for the statement in lines 61-62.

**Robotics Focus:**

Highly relevant to robotics but no hardware experiments

**Summary Of Paper:**

The co-design methods in the literature mainly consider discovering an optimal robot morphology and controller for a specific task. Yet, biological morphologies can perform multiple tasks. Drawing inspiration from pleiotropy in biology, this paper proposes a learning framework for designing a modular soft robot capable of performing a wide variety of tasks. The authors achieve fast adaptation to new co-design tasks by leveraging past co-design knowledge and demonstrate the effectiveness of brain-body pre-training. The paper presents simulation results to evaluate the performance of the proposed approach.

**Summary Of Recommendation:**

Motivated by the idea of facilitating effective body-environment interaction, the paper focus on co-designing both the physical bodies and the control systems that operate on the robot body. Their implementation of "pleiotropic co-design policy" is novel and can generate effective results against similar algorithms.

---

### Official Review · Reviewer_rtsy · 2023-07-20

**Confidence:** 5
**Originality:** Fair
**Technical Quality:** Good
**Clarity Of Presentation:** Very Good
**Impact:** 3

**Recommendation:**

Weak Accept: I recommend accepting the paper, but will not argue for my recommendation if the majority of other reviewers have a different opinion.

**Review:**

STRENGTHS
+ The paper is well written and easy to read.
+ The topic of the paper is highly relevant---the problem of joint optimization of robot design and control has garnered renewed interest in the robot learning community within the last few years.


WEAKNESSES
- The significance of the paper's contributions is unclear. It is not the first to propose using a universal controller/policy as part of joint optimization (e.g., Reference [33] in the paper and References [1-4] below) as a means of improving the efficiency and (in some cases) performance of the resulting design. Also relevant, many of these papers similarly motivate the idea of learning a single policy that can control different designs as a form of multi-task RL (e.g., [42] in the paper and [5, 6] below). Instead, the primary novelty lies in optimizing over a set of different tasks as a pre-training phase in an effort to facilitate "transfer" to the target task. The benefits are fairly clear in the context of zero-shot and few-shot transfer, but the results don't demonstrate that such an approach is better than training from scratch on the target task. However, it's not obvious whether there is anything fundamental to the algorithm that enables multi-task pre-training as opposed to training a transformer-based framework that has a universal policy on a set of different tasks. Indeed, there is no discussion of how one might go about deciding on these pre-training tasks when there is a target task in mind. It seems that they were chosen randomly.
- Related, Table 2 only includes one domain from Table 1. It is important to compare the performance of the resulting designs as a result of fine-tuning to those that are trained from scratch with the same architecture as well as a baseline (e.g., see the results in the "Evolution Gym" paper).
- The paper is missing a few references to relevant work in soft robot co-design [1, 2, 5, 7]. Reference [5] seems particularly relevant given the focus on soft robots and the use of a universal controller, though it seems that they only train on a single task.
- The paper doesn't provide sufficient details about the design space
- The paper provides no discussion of how one might extend these results to physical hardware, which is likely difficult given the significant sim-to-real gap, particularly for voxel-based designs and differentiable simulators [3].
- Certain claims in the paper are not supported. This includes the statement that the universal policy "significantly facilitates exploration of the design space". The results show that it improves performance including in a zero-shot setting, but there is no results that show improved exploration.
- Lines 148-150: The statement regarding deficiencies in optimization as performed by existing methods that alternate between updating the design and the control are not substantiated.
- While the appendix provides the hyper-parameter settings, it would be helpful if there was more discussion of the training procedure in the main paper, including the way in which tasks are chosen during pre-training (i.e., how often and from what distribution (uniform?)), how the method alternates between updating the design distribution and rolling out the current policy, etc.
- Personally, I find the frequent parallels to "brain" and "body", "genome", and "pleiotrophy" to be a distraction since the relationship between biological evolution and what the paper describes is purely superficial.
- Minor: Section 1 (first paragraph): After stating the importance of co-design, the motivating examples that are provided demonstrate the importance of optimizing the design, but not control.
- Minor: Lines 240-241: Any claims about why the model learned a particular design are purely speculative


GRAMMAR:
* Line 187: "Does our method, PreCo, ~~can~~ effectively perform"


REFERENCES

[1] R. Deimel, P. Irmisch, V. Wall, and O. Brock. Automated co-design of soft hand morphology and control strategy for grasping. In Proceedings of the IEEE/RSJ International Conference on Intelligent Robots and Systems (IROS), 2017.

[2] S. Kriegman, S. Walker, D. Shah, M. Levin, R. Kramer- Bottiglio, and J. Bongard. Automated shapeshifting for function recovery. In damaged robots. in Proceedings of Robotics: Science and Systems (RSS), 2019.

[3] S. Kriegman, A. M. Nasab, D. Shah, H. Steele, G. Branin, M. Levin, J. Bongard, and R. Kramer-Bottiglio. Scalable sim-to-real transfer of soft robot designs. In Proceedings of the IEEE International Conference on Soft Robotics (RoboSoft), 2020.

[4] D. Pathak, C. Lu, T. Darrell, P. Isola, and A. A. Efros. Learning to control self-assembling morphologies: A study of generalization via modularity. In Advances in Neural Information Processing Systems (NeurIPS), 2019.

[5] C. Schaff, A. Sedal, and M. R. Walter. Soft Robots Learn to Crawl: Jointly Optimizing Design and Control with Sim-to-Real Transfer. In Proceedings of Robotics: Science and Systems (RSS) 2022

[6] C. Schaff, and M. R. Walter. N-LIMB: Neural Limb Optimization for Efficient Morphological Design. arXiv:2207.11773 2022

[7] A. Spielberg, A. Amini, L. Chin, W. Matusik, and D. Rus. Co-learning of task and sensor placement for soft robotics. IEEE Robotics and Automation Letters, vol. 6, no. 2, pp. 1208–1215, 2021

**Quality Of The Limitations Section:**

Limitations are not well addressed

**Questions For Rebuttal:**

1) Can the authors clarify the nature of the time index for $s_t^d$ and $s_t^c$. In the case of the design state $s_t^d$, my understanding is that $t$ indexes the point in training, whereas in the case of the controller state $s_t^c$ it is episode time. Is that correct?

2) Line 180: What is meant by "At every time step, $K$ state information of each co-design task is uniformly sampled"? What is $K$? What is the "state information of each co-design task"?

3) How does fine-tuning on the target task affect performance on the training tasks?

**Robotics Focus:**

Relevant but unlikely to deploy to hardware in near future

**Summary Of Paper:**

The paper describes a framework (PreCo) for the joint optimization (co-design) of the voxel-based design and corresponding control policy for soft robots. Integral to the approach is its use of a pre-training phase to jointly optimize design and control on a set of training tasks before fine-tuning on a different target task. The policy is shared across designs and both the design and control are trained via model-free RL. Experiments demonstrate that PreCo outperforms curriculum-learning and meta-learning baselines in terms of zero-shot generalization to the target task as well as few-shot (with the exception of one domain).

**Summary Of Recommendation:**

The problem of co-optimization is interesting and relevant to robot learning and robotics, more generally, The core contribution is the use of a pre-training phase in which the framework performs joint optimization on a set of tasks as a means of facilitating zero- and few-shot transfer to a target task. However, it is not obvious whether there is anything fundamental to the architecture that enables this transfer vs. perhaps being the first to try a transformer-based universal control policy (which is not new) with a series of tasks. Without key insights into how these tasks should be chosen, the significance of the contributions is unclear.

UPDATE AFTER AUTHOR RESOPNSE/DISCUSSION

The authors' response addresses the primary concerns that I had with the paper and as a result, I have increased my score.

---

### Author Response · Authors · 2023-08-14
**[To All] Brief Summary of Rebuttal (Part 1/3)**

We would like to extend our heartfelt gratitude to all reviewers for their insightful and comprehensive feedback. We're pleased to note that the reviewers found:

(1)	Our research topic is interesting and highly relevant to the domain of co-design in robotics (all reviewers).

(2)	Our proposed method, PreCo, is an innovative and sound technology for designing and controlling modular soft robots to perform multiple tasks (reviewer WFDJ, Bcvj).

(3)	Our experimental investigations are impressive (reviewer dKrr, Bcvj) and our work has substantial contributions to the domain of robot co-design (reviewer Bcvj).

In response to the comments provided by reviewers, we have refined the manuscript, eliminated any ambiguities or potentially misleading phrasing, added more implementation details and conducted further experiments and in-depth performance evaluations to show the advantages of our method. Below, we detail the principal modifications:

**Clarifications and discussions**

(1)	We clarified our motivation for the selection of pre-training co-design tasks in Section 4 and Section 5.1 (reviewer rtsy, Bcvj).

(2)	We added more descriptions and implementation details of PreCo in Section 4 and Appendix B (reviewer rtsy, Bcvj).

(3)	We expanded the discussion on our paper's limitations and the potential adaptation of our method to physical hardware in Section 6 and Appendix D (reviewer rtsy, dKrr and Bcvj).

**Experiments**

(1)	Inspired by reviewer rtsy and dKrr, we added new baselines and conducted experiments to show that PreCo is better than those methods which learn from scratch, and the pleiotropic structure benefits the exploration (Section 5.2).

(2)	We conducted a pre-training experiment that encompasses a more diverse range of co-design tasks (reviewer Bcvj).

(3)	We experimented to investigate how fine-tuning the target task affects performance on the pre-training tasks under the co-design setting (reviewer rtsy).

**Our video for the rebuttal phase is available here:** https://files.catbox.moe/nubqw5.mp4

Kindly inform us if there are any further comments, inquiries, or areas of concern. Once again, we deeply appreciate the invaluable efforts and dedication of all reviewers in enhancing our work.

---

> ### Author Response · Authors · 2023-08-14
> **[To All] Response to common concerns (Part 2/3)**
>
> **The selection of pre-training tasks**
>
> Regarding the selection of pre-training tasks, we would like to clarify that they were not chosen randomly. For effective brain-body pre-training, we operate under the assumption that the target tasks need to share structural similarities with the pre-training tasks. Such an assumption is foundational to most transfer learning or pre-training methodologies.
>
> On the one hand, the target tasks used in our study essentially represent varying adaptations of the pre-training tasks. These target tasks include the following types:
>
> 1.	More challenging scenarios, such as the transition from ObstacleTraverser-v0 to ObstacleTraverser-v1 where the terrain becomes increasingly uneven.
>
> 2.	Transfers of comparable difficulty, as seen when moving from Walker-v0 to BridgeWalker-v0 (with a shift to softer terrain) or from PlatformJumper-v0 to GapJumper-v0, wherein the gap between steps expands but the height of these steps reduces.
>
> 3.	``Reverse" scenarios, exemplified by the transition from UpStepper-v0 to DownStepper-v0, where the direction of the steps is inverted.
>
> On the other hand, if a target task requires the robot to master a complex skill, such as traversing across extremely uneven terrain (e.g., ObstacleTraverser-v1), this skill can be broken down into some foundational abilities like walking, ascending stairs and surmounting minor obstacles. During pre-training, the universal co-design policy aims to extract basic brain-body links from these tasks and merge them. When facing specific target tasks, the policy is anticipated to leverage the prior knowledge, thereby alleviating the co-design challenge.

---

> > ### Author Response · Authors · 2023-08-14
> > **[To All] Response to common concerns (Part 3/3)**
> >
> > **The discussion of real-world experiments**
> >
> > Although in our paper, we tested our method using a simulator with relatively fundamental modules as a proof of concept to show its effectiveness, physical experiments will definitely be our next step. We discuss this sim-to-real issue in the following context.
> >
> > From the perspective of ``Sim”, the EvolutionGym platform [1] used in our work employs several simplifications to reach a trade-off between the simulation quality and velocity. Thus, for more realizable sim-to-real transfer, an improved version of its physics engine is needed to model soft-body physics in 3D space. We believe that using the Finite Element Method (FEM) numerical simulation would be one of the feasible ways to narrow this sim-to-real gap because the voxel-based design provides a naturally organized mesh, and its resolution and element type could be relatively easily determined. Moreover, FEM has shown promise for real-world soft robotic models when combined with highly stable implicit Euler integration [2].
> >
> > In our study, we focus on the model-free co-design problem where system modeling is not required, and we use reinforcement learning to approximate the gradient of design and control. This endeavor requires a large amount of training data. However, when considering the real-world problem, it is possible to transfer to the model-based setting. By leveraging the differentiable properties of certain FEM models [3], the universal parameterized co-design policy can be updated using Back-Propagation Through Time (BPTT), resulting in more efficient brain-body pre-training. When we move to the 3D modular robot design space, we would like to adapt the parameterization method depicted in Figure 3 to its 3D version. By using a ``3D convolutional kernel”, we can easily create input design observation sequences for the transformer-based policy. Although transformer has sufficient capacity to handle long sequences, a more expansive design space (e.g., thousands of voxels) would necessitate a more complex transformer model. This, in turn, could demand significantly higher computational power.
> >
> > From the perspective of ``Real”, one of the foremost considerations is the material selection, which should be predicated upon the required flexibility, durability, and functionality [4]. With this criterion, the Diels-Alder (DA) polymer [5] or silicone voxels, in conjunction with multi-material cubic blocks produced through 3D printing, may serve as ideal components for constructing the body of a modular soft robot. To establish the local observation space, each voxel could be equipped with an array of sensors, such as touch, pressure and velocity sensors. Alternatively, soft sensors, crafted from conductive elastomers that alter resistance upon deformation, could offer valuable feedback to the control system. Furthermore, Peano-HASEL actuators [6] or pneumatic actuators might be suitable for volumetric actuation (probably limited to expansion for efficient simulation) and closed-loop control could be achieved by utilizing Neural Networks (NNs). In the real-world setting, factors like material imperfections, air resistance, friction and many others come into play, we also need to iteratively refine the design and control algorithms based on real-world feedback.
> >
> > We acknowledge that each point discussed above presents its challenges but is well worth in-depth investigation, and we aspire for our work to serve as a catalyst for future research into the co-design of modular soft robots.
> >
> > **More implementation details of PreCo**
> >
> > We have added more details of our unified state-action space, network implementation, training procedure and baselines in Section 4, Section 5.1, and Appendix B.
> >
> > **References**
> >
> > [1] J. Bhatia, H. Jackson, Y. Tian, J. Xu, and W. Matusik. Evolution gym: A large-scale benchmark for evolving soft robots. In NeurIPS, 2021.
> >
> > [2] M. Dubied, M. Y. Michelis, A. Spielberg, and R. K. Katzschmann. Sim-to-real for soft robots using differentiable fem: Recipes for meshing, damping, and actuation. IEEE Robotics and Automation Letters, 7(2):5015–5022, 2022.
> >
> > [3] E. Nava, J. Z. Zhang, M. Y. Michelis, T. Du, P. Ma, B. F. Grewe, W. Matusik, and R. K.Katzschmann. Fast aquatic swimmer optimization with differentiable projective dynamics and neural network hydrodynamic models. In International Conference on Machine Learning, pages 16413–16427. PMLR, 2022.
> >
> > [4] M. Tebyani, A. Spaeth, N. Cramer, and M. Teodorescu. A geometric kinematic model for flexible voxel-based robots. Soft Robotics, 10(3):517–526, 2023.
> >
> >  [5] J. Legrand, S. Terryn, E. Roels, and B. Vanderborght. Reconfigurable, multi-material, voxel-based soft robots. IEEE Robotics and Automation Letters, 8(3):1255–1262, 2023.
> >
> >  [6] N. Kellaris, V. Gopaluni Venkata, G. M. Smith, S. K. Mitchell, and C. Keplinger. Peano-hasel actuators: Muscle-mimetic, electrohydraulic transducers that linearly contract on activation. Science Robotics, 3(14):eaar3276, 2018.

---

### Decision · Program_Chairs · 2023-08-30

**Decision:**

Accept (Oral)

**Comment:**

The paper concerns a novel idea, and provides some working solutions to a long-held issue with robotic design.

The authors have been responsive during the review phase.

It is important to put this work in context of the large body of work on co-optimization that dates back several decades and that has received renewed attention within the last several years.

Furthermore, the biological framing for this work is unwarranted as any biological comparisons are superficial at best, and misleading at worst.